On the elephant trails: habitat suitability and connectivity for Asian elephants in eastern Indian landscape

Palei Himanshu Shekhar 1
Jangid Ashish Kumar ashishjangid22@gmail.com 2 3
Hanumant Dhamdhere Dhanraj 4
Palei Nimain Charan 5
Mishra Arun Kumar 6
1 Aranya Foundation , Bhubaneswar , Odisha , India
2 Bisalpur , Pali , Rajasthan , India
3 Current affiliation: Wildlife Institute of India , Dehradun , Uttarakhand , India
4 Keonjhar Forest Division, Divisional Forest Office , Keonjhar , Odisha , India
5 Department of Zoology, North Orissa University , Baripada , Odisha , India
6 Regional Chief Conservator Office , Rourkela , Odisha , India
Sunny Armando
Electronic publication date: 2024 Mar 29
Publication date: 2024
Volume: 12
Electronic Location ID: e16746
Received 2023 Sep 5; Accepted 2023 Dec 11
Copyright: ©2024 Palei et al.
Copyright year: 2024
Copyright holder: Palei et al.
License: This is an open access article distributed under the terms of the Creative Commons Attribution License, which permits unrestricted use, distribution, reproduction and adaptation in any medium and for any purpose provided that it is properly attributed. For attribution, the original author(s), title, publication source (PeerJ) and either DOI or URL of the article must be cited.
License URL: https://creativecommons.org/licenses/by/4.0/

Keywords: Corridor, Ensemble spatial model, Habitat linkage, Protected areas, Landscape management

Funding: The authors received no funding for this work.

==============================
Identifying suitable habitats and conserving corridors are crucial to the long-term conservation of large and conflict-prone animals. Being a flagship species, survival of Asian elephants is threatened by human-induced mortality and habitat modification. We aimed to assess the habitat suitability and connectivity of the Asian elephant Elephas maximus Linnaeus, 1758 habitat in the state of Odisha in eastern India. We followed the ensemble of spatial prediction models using species presence data and five environmental variables. We used least-cost path and circuit theory approaches to identify the spatial connectivity between core habitats for Asian elephants. The results revealed that normalized difference vegetation index (NDVI; variable importance 42%) and terrain ruggedness (19%) are the most influential variables for predicting habitat suitability of species within the study area. Our habitat suitability map estimated 14.6% of Odisha’s geographical area (c. 22,442 km2) as highly suitable and 13.3% (c. 20,464 km2) as moderate highly suitable. We identified 58 potential linkages to maintain the habitat connectivity across study area. Furthermore, we identified pinch points, bottlenecks, and high centrality links between core habitats. Our study offers management implications for long-term landscape conservation for Asian elephants in Odisha and highlights priority zones that can help maintain spatial links between elephant habitats.

Introduction

Large herbivores play a crucial role in maintaining ecosystem structure (Bakker et al., 2006; Duncan et al., 2006). Unfortunately, their populations have been declining, which has disrupted the ecosystem dynamics (Asner et al., 2009). Globally, these animals face various threats, including habitat loss, fragmentation caused by human activities, conflicts with humans, and poaching (Ripple et al., 2015). Large herbivores maintain large home ranges and require vast habitat contiguity for survival, dispersal, and migration (Ceballos et al., 2005; Linnell et al., 2016). However, habitat fragmentation restricts their movement and prevents them to access suitable habitats. As a result, interlinked species and genetic diversity decline, increasing the risk of extinction (Berti & Svenning, 2020). Therefore, identification of suitable habitats and their connectivity is important for wildlife conservation in fragmented landscapes (Ghoddousi et al., 2021).

Conserving large herbivore populations in a complex of sustainable habitat requires creating a functional conservation network (Ripple et al., 2015). However, quantifying connectivity between habitat patches is challenging as species distribution, landscape resistance, and dispersal ability are unknown for majority of species. Prediction of suitable habitats based on species presence (or presence/absence) has been growing in various conservation research and is often valuable in conserving threatened species (Guisan, Thuiller & Zimmermann, 2017). It can be achieved through statistical models based on known species locations and their preferred environmental patterns (Guisan & Thuiller, 2005; Phillips et al., 2009). One such popular approach is species distribution modelling (SDM), which predicts suitable habitats and supports the conservation of threatened species, e.g., Asian slow loris (Nycticebus spp.) (Thorn et al., 2009). Recently, several algorithms have been developed to improve this approach (Guisan, Thuiller & Zimmermann, 2017). However, each algorithm has limitations, and relying on a single model can increase bias and uncertainty. Additionally, the model output of different algorithms may vary (Thuiller et al., 2019). By integrating the predictions of various modelling algorithms, ensemble models can reduce the biases and uncertainties associated with individual models and produce a more robust and reliable model (Thuiller et al., 2019).

Connectivity modelling often predicts connectivity using a resistance surface derived from a habitat suitability model. The habitat suitability surface is inverted to represent lower movement costs in areas of higher suitability (Mateo-Sánchez, Cushman & Saura, 2014). Various data types are used to study landscape connectivity, including species presence data, telemetry data, genetic data and expert opinion (Rezaei et al., 2022). Also, landscape connectivity has been quantified through various methods, including landscape genetics, graph-theoretic methods, least-cost paths, and circuit theory (Cushman et al., 2013; Thatte et al., 2021). The most commonly and widely applied ones are the least-cost path model and circuit theory (Cushman et al., 2013). The least-cost path represents the pathway with the lowest accumulated movement cost between two habitat patches. It assumes that animals disperse based on their knowledge of the surrounding landscape (Huck et al., 2011). On the other hand, circuit theory, similar to an electric circuit, assesses all potential pathways between patches, identifying areas with high conductance (/low resistance). It assumes that dispersers have information only about their immediate environment (McRae et al., 2008).

Globally, Asian elephants (Elephas maximus Linnaeus, 1758) are among the most threatened large herbivores (Shaffer et al., 2019), considered flagship and umbrella species due to their vulnerability to anthropogenic threats, coexistence with other threatened species, and requirement of large areas to survive (Sukumar, 2006). With approximately 50,000 individuals left in the wild, Asian elephants have experienced a decline of 95% in their historical range, and the remaining range is highly fragmented (Sukumar, 2006; Williams et al., 2020). India holds the largest population of Asian elephants, estimated at 60% of the global population, occupying over 110,000 km2 of diverse habitats ranging from moist evergreen forests to semi-arid scrub, grasslands, and human-modified landscapes (Sukumar, 2006; Goswami & Vasudev, 2017). However, the large-scale modifications due to the growing human population and rapid economic growth come with the development of associated infrastructure, therefore, urbanization and agriculture expansion fragment the habitat and increase the isolation between remnant population of Asian elephants (Sukumar, 2006; Goswami & Vasudev, 2017; Shaffer et al., 2019). A recent genetic study in India confirms that the presence of natural and anthropogenic barriers affects the genetic connectivity of Asian elephants (De et al., 2021).

Considerable studies have been conducted on the habitat suitability and connectivity modelling of the Asian elephant (Puyravaud et al., 2017; Suksavate, Duengkae & Chaiyes, 2019; Vasudev et al., 2021; Wilson et al., 2021; Neupane et al., 2022). These studies indicate that several ecogeographical (/environmental) factors are associated with habitat suitability and connectivity. For example, Neupane et al. (2022) reported the habitat suitability of Asian elephants is influenced by altitude and precipitation in Nepal. Vegetation cover, proximity to forests, and human disturbance (population density) determined the habitat suitability of Asian elephants in Northeast India (Vasudev et al., 2021). Elephants also preferred lower elevation, gentle slopes, and proximity to water, whereas they avoided rugged terrain and roads (Suksavate, Duengkae & Chaiyes, 2019; Sharma et al., 2020; Wilson et al., 2021). Puyravaud et al. (2017) applied the UNIversal CORridor (UNICOR) connectivity modelling tool to predict optimal movement corridors across the region based on the least-cost routes derived from the resistance map in southern India. Vasudev et al. (2021) used the randomised shortest path (RSP) framework to model connectivity in northeast India. Chaiyarat et al. (2023), Suksavate, Duengkae & Chaiyes (2019) and Neupane et al. (2022) utilized least-cost path (LCP) and circuit theory approaches to quantify the connectivity between core habitats of elephant populations in Thailand and Nepal, respectively. Despite these studies, more studies are needed to cover the specific information from various landscapes within its range and to elucidate the effects of different factors on habitat suitability and connectivity of Asian elephants. It will facilitate the reconciliation of development projects and Asian elephant conservation while providing scientific guidance for landscape-specific targeted conservation initiatives.

In this study, we analysed the habitat suitability and connectivity between core habitats of the Asian elephant in Odisha using species occurrence data. We used an ensemble of multiple SDM approaches to analyse these data. Approximately 2,000 Asian elephants persist in Odisha’s fragmented, human-modified landscape, which is part of the central Indian population. Recently, a study found genetic evidence for the connectivity of Asian elephant populations between the northern and central parts of Odisha (Parida et al., 2022). Therefore, it is important to understand the possibility of movement between these populations to design and implement management actions and identify possible threats to the long-term survival of the species. Thus, our objectives were to (1) evaluate state-wide habitat suitability with respect to environmental variables, and (2) map connectivity between core habitats using the least-cost path and circuit theory models.

Material and Methods

Study area

Odisha state is situated in the eastern part of India (81°24′–87° 29′E and 17°48′–22°34′N), covering an area of 155,707 km2 (Fig. 1). Affected by the Southwest and Northeast monsoons, the state bears an annual mean temperature of 15 °C–38 °C and annual precipitation of 960–1,870 mm. During the monsoons, the humidity level is high (c. 80%) for four months (July to October). Summer (March to June) and winter (November to February) are the other two seasons. The state has 51,345 km2 of forest cover, represents ∼33% of its geographical area (India, 2017). The forest types of Odisha are mainly categorized into tropical moist deciduous, tropical dry deciduous, and a few areas of semi-evergreen forests (Champion & Seth, 1968). Odisha can be divided into three major physiographical regions: the middle mountainous region, the western rolling uplands, and the eastern coastal plains (Sinha, 1971). The mountainous region constitutes approximately three-quarters of the state. Generally, the topography is undulating, except for the high plateaus and coastal plain areas. The mountainous region of the state is one of the most mineral-rich areas of the country. Around 60% of the state’s population is dependent on agriculture, and small-scale cultivation is the predominant livelihood strategy of the people (Panda & Padhi, 2021). Rice is the primary crop in the state, grown during the monsoon season (July to October) and harvested between October and December (Hoda et al., 2021). The state has a population density of 269 per km2 and 41.9 million people; primarily (80%) live in rural areas. Also, the state has the third-largest tribal population in the country, accounting for 22.8% of the total population of the state (Dungdung & Pattanaik, 2020).

Figure 1 The map showing Odisha state with the spatially thinned presence events (n = 5,568) of Asian elephants along with the overlay of forest cover and nightlight information.

Map created by Ashish Kumar Jangid, using ArcMap version 10.5 software.

The state has 19 Wildlife Sanctuaries, one National Park, two tiger reserves, and one biosphere reserve, constituting 4.73% of the state’s geographical area. Additionally, the state has also declared three elephant reserves, namely, Mayurbhanj Elephant Reserve, Mahanadi Elephant Reserve, and Sambalpur Elephant Reserve. The Odisha state has 30 administrative districts and 50 forest divisions.

Asian elephant presence data, spatial filtering and spatial shift of conflict events

We collected presence data (n = 6,352) of the Asian elephants through direct observations and newspaper reports between January 2011 and January 2022. The direct observations included direct sightings, indirect signs of the species (i.e., footprint, dung and feeding signs) through opportunistic surveys of Asian elephants or other large mammals, and human-elephant conflict incidences (attack on humans and livestock, raid crops and damaging houses or other infrastructures) in different protected areas (PA) and forest divisions. The opportunistic direct observations of Asian elephants were obtained from various sources, including incidental observation, transect walk and forest road survey. Data were collected without following a systematic approach across state forest divisions in protected and unprotected areas of forest patches. During transect or forest road surveys, two observers who were familiar with Asian elephant signs have recorded the location. Our survey yielded 2,432 direct observation and 195 conflict locations covering nine protected areas and 22 forest divisions across the state.

Our opportunistic survey may be prone to geographic bias as it may not have sampled the full range of environmental conditions in which the species occur (Phillips et al., 2009; Yackulic et al., 2013; Guillera-Arroita et al., 2015). Therefore, to understand the presence of Asian elephants across the state, we also collected information on Asian elephants daily from regional Odia newspapers from 2017 to 2021. We selected two newspapers due to their wide reach throughout the state and availability of online publishing of local news. In addition to elephant presence information, we also collected geographical information such as the name of the reported village, Gram Panchayat, Tehsil, and district from the newspapers. We projected geospatial coordinates of elephant incidents referring the centroid of the reported village as an approximate location of the elephant presence. We derived the digitized village boundary map from the Survey of India. The average area of villages was 2.92 km2, and the mean distance between centroid and village boundary was 0.44 ± 0.25 SD km, which indicates a maximum error of <1 km in the projected presence locations. Conflict records involving house damage, crop raid or human injury and death by Asian elephants were also collected at the same time from newspaper and location were manually filtered to avoid duplicate records. Given the absence of other sympatric large-sized species, its presence and signs are easily detected and identified; consequently, the presence data is highly accurate. We yielded 3,725 presence records of elephants from newspapers covering 43 forest divisions and assumed that all potential elephant habitats were adequately covered. We pooled the elephant locations from direct observations and newspapers from 2011 to 2022 to create a comprehensive dataset.

Several studies indicated that human-elephant conflict incidents are often associated with the proximity of human settlements to forested areas (Sukumar, 1993; Gubbi, 2012; Neupane et al., 2019). Therefore, we also assumed that any conflict-caused elephant in or around the village area was moved in from the nearest forest patch. Subsequently, we spatially simulated and shifted 2,613 conflict events into the nearest forest patch using the “create random points” tool in ArcMap, and the shift distance opted randomly between 5–10 km selected based on home range and daily movement of Asian elephants (Fernando et al., 2008; Baskaran, Kanakasabai & Desai, 2018; Torre et al., 2019). The spatial shift of presence events was intended to model the habitat connectivity for elephants in Odisha rather than spatial trends of conflict. However, our opportunistic surveys and newspaper reports may bias toward easily accessible sub-optimal habitats that have been more extensively reported than other habitats. Therefore, to reduce the sampling bias on model performance and to avoid autocorrelation, we spatially filtered presence locations on the 1 km spatial resolution using the “Spatially Rarefy Occurrence Data for SDMs” tool under SDMtoolbox version 2.4 on ArcMap version 10.8. This procedure resulted in 5,568 spatially independent species presence records from 45 forest divisions (114.06 ± 13.8 SE records/division, 32–492) and 13 protected areas (36.93 ± 8.04 SE records/PA, 2–113) across the state. These presence-only locations were further considered for spatial analysis.

Environmental variables selection

An extensive array of non-correlated environmental variables was selected (r<—0.6—) (Riley, De Gloria & Elliot, 1999; Rood, Ganie & Nijman, 2010; Thapa, Kelly & Pradhan, 2019; Sharma et al., 2020; Wilson et al., 2021; de la Torre et al., 2021). Variables included average annual precipitation (bio12), distance from nearest protected areas, elevation, terrain ruggedness index, and normalized difference vegetation index (Fig. S1; Table S1). Annual precipitation (mean of annual precipitation for the year 2000-2017) was used to represent the vegetation growth and to distinguish coastal areas. This information was acquired from the WorldClim database version 2.0 (http://www.worldclim.org; Fick & Hijmans, 2017). Protected area boundary (includes tiger reserves, National Parks, and wildlife sanctuaries) was obtained from the GeoNode online portal (https://geo.ejatlas.org) and cross-validated with PA map of state forest department. Elephants utilize many of these protected areas throughout the year as a refuge or halt sites while migrating. Therefore, we used the “Euclidean Distance” tool in ArcMap version 10.8 to calculate the shortest distance from the protected area. Elevation data (SRTM 250 m) was downloaded from USGS earth explorer (http://earthexplorer.usgs.gov). Terrain ruggedness index was computed using “Terrain Ruggedness Index (TRI)” tool in QGIS version 3.22.1 Białowieża from elevation data (Riley, De Gloria & Elliot, 1999). Normalized difference vegetation index (NDVI) was downloaded from the Terra-NDVI AppEEARS NASA earth data portal (http://appeears.earthdatacloud.nasa.gov) from 2011 to January 2022. NDVI data was acquired from the middle of January, April, July and October of every year from 2011 to 2021 to represent local seasons. Mean NDVI was pooled from all 45 NDVI layers using the “Raster Calculator” tool in ArcMap version 10.8, further considered one of the predictor variables. All the predictor variables were resampled to 1 km spatial resolution.

To evaluate multicollinearity among variables, we checked the pairwise Pearson correlation coefficient (r) between covariates and considered r =—0.6— as the threshold for variable selection (Khosravi, Hemami & Cushman, 2018). Furthermore, we tested the variance inflation factor (VIF) using package “usdm” version 1.1 in the R studio to sidestep the raster collinearity. We included variables for which the VIF<10, indicating least collinearity between variables (Guisan, Thuiller & Zimmermann, 2017).

Habitat suitability modelling

The habitat suitability model for the Asian elephant in Odisha was generated by ensemble modelling approach using biomod2 version 3.5.1 package in R studio version 1.4.1717 (Thuiller et al., 2021). The precise and more accurate ensemble model was produced with combining generalized linear model (GLM), generalized additive model (GAM), gradient boosted machine (GBM), maximum entropy (MaxEnt), multiple adaptive regression splines (MARS) and random forest (RF) algorithms (Araújo & New, 2007; Marmion et al., 2009). We also estimated the contribution of predictor variables to the ensemble model. Our response variable, i.e., binomial matrix of presence and pseudo-absence (one set of random selection) of the species, was modelled against the predictor variables (Phillips et al., 2009). Considering the optimum ratio, the number of pseudo-absences was kept double of independent presence records (n = 11, 136). We examined the predictive ability of mentioned algorithms using 10-fold cross-validation; 80% of response data was selected randomly for training and 20% for testing the prediction (Guisan, Thuiller & Zimmermann, 2017; Thuiller et al., 2019).

Model performance was tested using receiver operating characteristics (ROC; area under curve, AUC), the ratio of true positives among positive predictions against false positives among negative predictions. AUC = 0.5 indicates that the model’s performance is not better than random, while AUC closer to 1.0 indicates a better performance. Also, Cohen’s kappa (Heidke skill score; KAPPA) and true skill statistic (TSS) were measured for evaluating relative accuracy. The TSS values vary from −1 to +1, where +1 signifies perfect performance, and 0 or less than suggests random prediction. Further, we computed the weighted pool average of models according to their predictive accuracy as independently assessed (Thuiller et al., 2009). Model execution was dominantly executed using binomial logit and Bernoulli distributions. Predicted spatial probabilities were pooled, and the average of the better-fitted models was plotted with predictors for the response assessment. Models, which have >0.79 AUC (around ∼50% top fitted models), were selected for the ensemble (Jarnevich et al., 2018). The ensemble output depicts a relative gradient of habitat suitability for elephants across the study area and, hence, the probability of habitat suitability was rescaled to 0–100.

The habitat suitability was further classified into five equal intervals, i.e., <20% for “least potential,” 21–40% for “moderately low potential,” 41–60% for “moderate potential,” 61–80% for “moderately high potential” and >80% for “highly potential habitat”. Division-wise habitat suitability for Asian elephants was classified, as forest divisions are used as the protection and management unit by the forest department.

Identifying core habitat patches and connectivity modelling

To understand the landscape connectivity of Asian elephants, we selected 19 core habitat nodes, comprising 11 protected areas and eight non-protected areas (Table S2). We identified core habitat nodes, where elephants’ presence has been constantly recorded for last ten years after reviewing previous literature, evaluating empirical observations and consulting expert knowledge. Core habitat nodes were defined as forest patches larger than 100 km2, annual home range of elephants (i.e., ranging from 100 to 1,000 km2). Protected area boundaries were obtained from GeoNode online portal, while unprotected core habitat patches were delineated from the Survey of India Topo-sheets (http://onlinemaps.surveyofindia.gov.in). The minimum and conservative distance between nodes were kept at 10 km based on the daily movement of elephant, i.e., 3.8−4.1 km (Chan et al., 2022).

Elephants have high cognitive abilities, retaining valuable information about past locations and strategically navigating to meet their needs (Sukumar, 1993; Sukumar, 2006; Vasudev et al., 2021). Therefore, we opted for a combination of least-cost path and circuit theory tools within the Linkage Mapper program (McRae & Kavanagh, 2011) in ArcMap version 10.8 to identify connectivity between core habitats. As the analysis required a resistance layer, a permeability map was developed for Asian elephants to generate the connectivity models. The inverse of ensemble habitat suitability (/conductance) was generated as a resistance map. This approach assumes that habitat suitability proportionally relates to the animal movement (Mateo-Sánchez et al., 2015). We first defined the least-cost routes connecting all the core habitat patches pairwise to identify the possible link of spatial connectivity. The least-cost model assumes that animals disperse according to previous knowledge of the surroundings. The model estimates the shortest and most cost-efficient distance between the core habitat patches, accounting for resistance in the movement (Adriaensen et al., 2003). Based on the core habitat nodes and resistance surface, the linkage model was run in order to map the least-cost linkages between adjoining core areas. The least-cost path is the route of maximum efficiency between two areas as a function of the distance travelled (length) and the cost traversed (accumulated cost) (Etherington & Holland, 2013). The connectivity linkages were evaluated under the threshold buffer of 50 km from modelled movement pathways, as the continuation of largely unsuitable habitats restricts the elephant movements. We computed the Euclidean distances (EucD) and lengths of Least-cost pathways between nodes. Furthermore, we used the ratio of euclidean distance to the length of the least-cost path (EucD: LCP) to compare the linkage quality and tortuosity. The higher ratio of the former metric indicates difficulties in moving between the core habitat pairs relative to how close they are. Later metrics describe the average resistance encountered for the optimal linking path.

Further, we used Circuitscape through Linkage Mapper to identify important areas for habitat connectivity. Unlike least-cost path models, multiple dispersals can be assessed using the circuit-based approach, which is theoretically based on random walk theory (McRae et al., 2008). As elephants are reported to utilize a wide variety of land use cover, we targeted to model all possible connectivity linkages among the core habitat nodes. For each pair of core habitats, circuitscape calculates multiple connecting paths from an electrical resistance surface. Movement often occurs in areas with the least resistance or higher conductivity. Effective resistance decreases as the number of pathways between core habitats increases. We also calculated current flow centrality by “Centrality Mapper” in the programme Linkage mapper to identify the importance of individual core habitats and linkages to maintain the connectivity within the landscape (Carroll, McRae & Brookes, 2012). Centrality scores of nodes and pathways were classified using the equal break classification, as least, moderate and highly centralized (high centrality is a high priority and so on). If the number of pathways reduces, effective resistance around pathways increases, and movement restricts to pinch points in the landscape. Higher current density within the pinch points indicates the linkages which animals are more likely to use, and no alternative links are available in the surrounding landscape. We mapped pinch points by “Pinch point mapper” in Linkage mapper to find the critical bottlenecks in connectivity; ultimately, these areas require greater conservation attention for the potential connectivity on the larger landscape. Finally, the map of movement corridors was reclassified into three categories, least, moderate and high movement-prone areas, to represent the most essential areas for Asian elephant movements.

Results

Habitat suitability modelling

Of 60 replicated models, 30 were chosen as better-fit models (AUC>0.79). Ten models under RF algorithm, eight models under GBM, three under MARS, three under GAM, and six under MaxEnt algorithm were selected for further spatial ensembling. TSS score of 0.42−0.53 suggested good accuracy of all models (Allouche, Tsoar & Kadmon, 2006). Ensemble model displayed pooled AUC 0.94, KAPPA 0.73 and TSS 0.71 (Table S3). Model-wise AUC, KAPPA, TSS and their weightage for ensembling are given (Table S3). In the ensemble model, GLM had the lowest performance (of 76%) in predicting habitat suitability for Asian elephants in the study area, while the rest algorithms had higher contribution (79–81%).

For the ensemble model, NDVI was the most significant predictor to characterize the habitat suitability or use probability of the Asian elephant (variable importance/contribution 42%; Table S4, Fig. 2). In increase of NDVI, the probability of habitat use by Asian elephant increased. As the terrain ruggedness increased from plain to undulating terrain, the likelihood of habitat use by Asian elephants also increased (19%). The optimal range of elevation for the habitat use of Asian elephants was c. 50–500 m (17%). As the distance to protected areas increased, the likelihood of habitat use of the Asian elephant decreased (13%). The Annual Precipitation (bio12) had little positive contribution to the ensemble model (8%).

Figure 2 Plots showing the predicted responses between variables and potential habitat use by Asian Elephants in Odisha, India.

The ensemble model predicted that 14.6% (22,442.73 km2) of the state was estimated as a highly potential for Asian elephant use, followed by 13.3% (20,464.37 km2) moderately high potential, 11.34% (17,436.44 km2) moderately potential, 15.14% (23,273.95 km2) moderately low potential areas and the remainder as the least potential (Fig. 3). Out of 50 administrative forest divisions, 46 of which recorded the Asian elephant’s high presence probability (psuitability >0.6). Focusing on high and moderately high potential areas (Thapa et al., 2018) (Table S5), 40% of forest divisions had more than 500 km2 of high predicted probability area of habitat use for Asian elephants (psuitability >0.6) (Table S5). Of the total area of the PAs, 75.34% (5,481.84 km2) were predicted to be potential habitats for Asian elephants (where psuitability >0.6; Table S6).

Figure 3 Map showing the ensembled probability of habitat use by Asian elephants in the state of Odisha, India (numbers refer to Table S5).

Map created by Ashish Kumar Jangid, using ArcMap version 10.5 software.

Core habitat and connectivity

We considered 19 core habitat nodes (average area 644.88 ±143.41 (SE) km2, range 115.47 to 2,750 km2) and identified 58 potential corridors for the Asian elephant to disperse across the landscape (Fig. 4; Table 1 & Table 2). The average length of corridors (least-cost pathway) was 126 ± 73.38 (SD) km (13.55 to 310.84 km). The mean euclidean distance between core habitat patches was 95.74 ± 59.7 (SD) km (10 to 271.95 km).

Figure 4 Maps showing the importance of potential movement pathways and core habitat nodes along with the pinch point status for the potential movement areas for Asian elephants in Odisha, India (numbers refer to Table 1 and Table S7).

Map created by Ashish Kumar Jangid, using ArcMap version 10.5 software.

Table 1 Details on the core habitat nodes (protected and unprotected area cluster), mapped for modelling the potential movement pathways for Asian elephants within the state of Odisha, India, where “Node ID” represents the unique ID of core habitat node.

Node ID	Core habitat node	Centrality score	Centrality rank	Area of habitat node (km2)	
1	Similipal	43.72	moderate	2,750.00	
2	Jada	38.64	moderate	905.54	
3	Siddhamath	32.51	least	341.46	
4	Garjanpahar	28.02	least	431.88	
5	Debrigarh	37.38	least	423.69	
6	Lakhari valley	30.56	least	196.27	
7	Khalasuni	55.56	moderate	696.55	
8	Anantapur	42.2	least	234.45	
9	Kapilash	41.48	least	317.87	
10	Satkosia	77.07	high	1,440.2	
11	Subarnagir	51.01	moderate	474.2	
12	Sunabeda	38.67	moderate	1,256.45	
13	Hattigam	27.21	least	115.47	
14	Karlapat	50.35	moderate	171.58	
15	Kothagarh	51.02	moderate	699.53	
16	Barabara	36.57	least	868.53	
17	Dharamagarh	20.07	least	541.99	
18	Chandaka	41.12	moderate	182.68	
19	Kuldiha	18	least	232.83	

Table 2 Details on all potential movement pathways for Asian elephants connecting core habitat nodes within Odisha, where “From Node” and “To Node” represent both end extent of protected area clusters, Euclidean distance and length of Least-cost pathways indicate their respective length of the distances, and Centrality indicates the linkage importance of connecting pathway.

From node	To node	Euclidean distance (km)	Length of least-cost pathway (km)	Ratio of euclidean distance & LCP	Effective resistance	Centrality score	Centrality class	*Length outside forest areas (km)	
1	2	131.41	179.91	0.73	13.08	6.71	least	57.66	
1	3	66.31	113.88	0.58	12.07	8.60	least	79.72	
1	4	223.90	272.80	0.82	22.42	5.22	least	112.19	
1	7	158.72	193.24	0.82	12.99	6.55	least	77.17	
1	8	65.12	77.13	0.84	8.91	14.50	moderate	11.08	
1	9	60.48	107.27	0.56	14.15	9.87	moderate	43.62	
1	19	11.78	13.55	0.87	2.98	18.00	high	12.97	
2	3	42.53	47.80	0.89	8.68	10.14	moderate	33.94	
2	4	51.38	91.40	0.56	11.20	6.60	least	51.40	
2	5	81.28	110.56	0.74	16.50	5.33	least	78.76	
2	7	34.14	42.31	0.81	4.31	13.82	moderate	25.96	
2	8	91.50	110.94	0.82	8.97	6.19	least	49.89	
2	9	165.66	220.99	0.75	14.82	4.95	least	98.86	
2	18	184.02	231.19	0.80	18.85	5.54	least	78.15	
3	4	137.18	182.17	0.75	19.08	4.56	least	105.85	
3	5	159.36	186.72	0.85	23.06	4.48	least	116.65	
3	7	98.84	110.82	0.89	10.26	6.72	least	71.86	
3	8	94.27	105.06	0.90	10.06	7.47	least	86.06	
3	18	169.40	288.75	0.59	21.19	5.06	least	125.41	
4	5	7.77	25.06	0.31	12.88	12.06	moderate	22.82	
4	7	42.95	50.72	0.85	11.27	9.61	moderate	29.75	
5	7	36.81	51.94	0.71	14.45	5.94	least	39.50	
5	9	218.68	246.28	0.89	25.96	3.49	least	102.29	
5	10	101.01	118.56	0.85	17.78	6.88	least	82.87	
5	11	90.70	128.34	0.71	19.89	7.91	least	97.12	
5	12	134.21	234.51	0.57	34.18	7.06	least	137.78	
5	18	227.87	256.48	0.89	29.37	3.61	least	81.58	
6	10	108.65	120.50	0.90	10.66	11.76	moderate	47.12	
6	11	119.95	159.36	0.75	10.56	5.68	least	55.77	
6	15	49.83	56.71	0.88	6.07	16.19	moderate	36.21	
6	16	52.77	78.35	0.67	11.69	9.49	moderate	41.35	
7	8	60.66	72.10	0.84	8.02	8.15	least	20.46	
7	9	134.23	164.45	0.82	13.35	5.23	least	67.73	
7	10	46.17	54.38	0.85	5.22	22.21	high	23.50	
7	12	185.99	221.25	0.84	24.82	8.94	moderate	112.86	
7	18	146.42	174.65	0.84	16.92	5.95	least	47.02	
8	9	32.11	39.46	0.81	10.03	8.28	least	28.48	
8	10	46.73	60.57	0.77	7.38	14.41	moderate	29.93	
8	12	271.95	310.84	0.87	28.61	7.40	least	157.36	
9	10	59.51	65.76	0.90	10.05	16.65	moderate	34.93	
9	18	13.81	19.01	0.73	10.35	16.48	moderate	12.92	
10	11	35.04	45.38	0.77	4.88	26.57	high	12.76	
10	15	84.06	107.58	0.78	7.54	14.15	moderate	41.34	
10	16	49.20	72.48	0.68	6.43	11.84	moderate	30.78	
10	18	61.02	75.96	0.80	12.58	11.69	moderate	14.00	
11	12	128.15	174.39	0.73	20.03	7.43	least	85.90	
11	14	106.17	113.68	0.93	10.22	17.45	moderate	54.62	
11	15	59.80	74.83	0.80	6.16	10.97	moderate	38.76	
11	16	101.12	124.46	0.81	9.74	8.02	least	49.76	
12	13	25.30	120.09	0.21	82.30	11.02	moderate	59.33	
12	14	55.40	92.31	0.60	19.77	10.66	moderate	34.66	
12	15	122.51	164.03	0.75	19.38	6.82	least	72.21	
13	14	89.69	156.86	0.57	65.47	15.89	moderate	52.96	
13	17	125.73	210.24	0.60	114.84	9.50	moderate	105.44	
14	15	55.03	65.27	0.84	5.72	26.04	high	33.78	
14	17	124.01	150.52	0.82	67.26	12.64	moderate	102.29	
15	16	81.05	117.65	0.69	10.01	9.87	moderate	63.97	
16	18	33.62	46.26	0.73	14.31	15.93	moderate	35.45	
Notes.

* Outside forest areas include agricultural fields or nearby areas to human settlement, which has low or moderate forest cover (Forest cover adopted from FSI, 2017: http://www.fsi.nic.in/forest-report-2017).

Node centrality analyses gave insights into the highly centralized nodes, which have the most significant connections with other nodes. Centrality values for all habitat cores varied considerably, with a mean value of 40.06 ± 13.63 (SD) (18 to 77.07) for the entire network. The highest centralized node (>57.38 centrality score) was only Satkosia (10). The moderate centralized nodes were (>37.69 and <57.38 centrality score) Similipal (1), Jada (2), Khalasuni (7), Anantpur (8), Kapilash (9), Subarnagir (11), Sunabeda (12), Karlapat (14), Kothagarh (15) and Chandaka (18), and the remaining eight found showing least centrality (<37.69 centrality score) (Table 1; Fig. 4).

The mean path centrality score was 10.18 ± 5.2 (SD) (3.49 to 26.57) for the entire network. The path centrality analysis suggested that four paths are highly centralized (>15.39 centrality score), 25 paths are moderately (>7.69 and <15.39) and rest 29 paths are found to be least centralized (<7.69 centrality score) (Table 2; Fig. 4).

Pinch point mapping provided insights on the forest areas connecting Similipal-Jada-Siddhamath-Khalasuni-Anantpur-Satkosia (1-2-3-7-8-10) and Satkosia-Subarnagir-Karlapat-Kothagarh (10-11-14-15) (Fig. 4), as these paths are surrounded by degraded land, densely populated human settlements, and agricultural fields. Rest corridors have broad areas facilitating the species’ movement throughout the state.

Discussion

Our results emphasized that NDVI, terrain ruggedness, elevation, and distance to protected areas are influential variables predicting the potential habitat use by Asian elephants. Studies have suggested that NDVI may not accurately reflect resource availability or habitat quality for elephants in tropical forests (Lakshminarayanan et al., 2016; Gautam et al., 2019). In our study, NDVI was positively correlated with Asian elephant habitat use, indicating a preference for diverse and abundant vegetation. Similar preferences where Asian elephants consistently select dense vegetation with different phenology were reported in the Asian elephant’s range (Thapa, Kelly & Pradhan, 2019; Bakri, Setiawan & Winarno, 2022; Amorntiyangkul et al., 2022). Several studies reported that elephants favour nutrient-rich lowland forest settings, including riparian forests and mountain valleys (Hedges et al., 2005; Pradhan & Wegge, 2007; Rood, Ganie & Nijman, 2010). However, our study found that the Asian elephants extensively use rugged landscape; possibly due to large forest areas available in rugged terrain and flat-less rocky areas are predominantly occupied by human habitations and agricultural land, contrary to previous studies. Wilson et al. (2021) shared similar results and suggested Asian elephants were forced to use non-preferred habitats due to human activities in flat land for developing settlement and agriculture. Moreover, depressions in rugged terrain are natural waterways that provide water and natural movement paths for Asian elephants for a longer time span (Pan et al., 2009; Shannon et al., 2009; Rood, Ganie & Nijman, 2010). Several studies suggested that the spatial and temporal availability of food and water resources are key factors in habitat selection by Asian elephants Kumar, Mudappa & Raman, 2010; Alfred et al., 2012; Taher et al., 2021). According to Thapa, Kelly & Pradhan (2019), elephants can prefer areas with high ruggedness due to the availability of forage and water. Similarly, it resulted in Asian elephants increasingly using rugged terrain in our study area too. Elevation, as another key contributor to our ensemble model, an altitude of 400 to 600 m was the optimum habitat for Asian elephants in our study, which corroborates the result of Areendran et al. (2011) and Sharma et al. (2020). These elevation ranges are often associated with forest cover and mountain valleys (Pan et al., 2009; Rood, Ganie & Nijman, 2010), which facilitate the suitable habitats for elephants. The reduced probability of Asian elephant habitat use in higher elevations reflects their avoidance of using such areas due to high-energy foraging costs (Wall, Douglas-Hamilton & Vollrath, 2006). However, our habitat suitability model predicted Similipal Tiger Reserve (average elevation of c. 850 m) as one of the most potential elephant habitats, where elephant presence was recorded up to 1,100 m. Our finding shows high habitat suitability in the proximity to protected areas and decreased with increasing distance from the protected area, possibly due to the high degree of protection status in and around protected areas (Amorntiyangkul et al., 2022; de la Torre et al., 2021).

The wide suitable range of these predictor variables indicates the generalist habit of Asian elephants. Our findings predicted the use of degraded multi-use forest patches interspersed with croplands by elephants (e.g., in Keonjhar forest division, Angul forest division and Balasore wildlife division). These low-quality habitats provide refuge for Asian elephants to raid crops and human settlements in the proximity (Calabrese et al., 2017; Tripathy et al., 2021; de la Torre et al., 2021). In Peninsular Malaysia, disturbed human-dominated agricultural landscapes are prime elephant habitats and have become human-elephant conflict hotspots (de la Torre et al., 2021). The occurrence of elephants in human-dominated areas may alter their behavior, like increasing aggressiveness due to greater human presence (Sukumar, 1993). Also, these areas have high rates of human-caused mortality, including retaliatory killing, poaching and electrocution (Palei et al., 2014).

Cumulative current flow models indicate linkage zones for maintaining a network of habitat patches in the study area. Among core habitat nodes, centrality scores indicated that Satkosia (Node 10), Similipal (1), Jada (2), Khalasuni (7), Anantpur (8), Kapilash (9), Subarnagir (11), Sunabeda (12), Karlapat (14), Kothagarh (15) and Chandaka (18) are important habitats with more than one linkage connecting them. Multiple linkages give animals alternative pathways and greater flexibility to move across the landscape. In recent years, Asian elephants’ presence has increased in Hattigam (13) and Dharamagarh (17) of south Odisha (Himanshu Palei, pers. obs., 2018). However, low connectivity is predicted in these corridors, namely, Debrigarh-Sunabeda (5–12), Siddhamath-Debrigarh (3–5), Siddhamath-Garjanpahar (3–4), Similipal-Jada (1–2), Subarnagir-Sunabeda (11–12), Lakhari Valley-Subarnagir (6–11), Debrigarh-Subarnagir (5–11), Subarnagir-Barabara (11–16), Debrigarh-Satkosia (5–10), Similipal-Siddhamath (1–3), Jada-Anantpur (2–8), Siddhamath-Khalasuni (3–7), Jada-Debrigarh (2–5) and Siddhamath-Anantpur (3–8) as these are long distance, less centralized, and single pathways surrounded by human activities. The linkages between Similipal-Kuldiha (1–19), Khalasuni-Satkosia (7–10), Satkosia-Subarnagir (10–11) and Karlapat-Kothagarh (14–15) have the existence of pinch bottlenecks and hold high centrality. The current flow between the large core habitat nodes, including Similipal (1) and Satkosia (10) demonstrates how intermediary habitat patches (Anantpur 8 and Kapilash 9) help the connectivity between long-distance habitat patches. Similipal and Satkosia are separated by a long distance, suggesting that comparatively smaller habitat nodes (Anantpur and Kapilash) are vital ‘stepping stones’ and can help for intergenerational gene flow between them. A recent study supports the gene flow between Similipal, Satkosia, and Chandaka, indicating that these linkages are potentially functional for Asian elephants (Parida et al., 2022). The paths linked on the state borders have suggested the interstate connectivity of elephant habitats with Odisha, e.g., forest areas around Similipal (node 1) and Siddhamath (3) are linked to Jharkhand. Also, the Similipal facilitates the pathways to the West Bengal state. Forest areas around Garjanpahar (4), Debrigarh (5), Sunabeda (12), Hattigam (13), and Dharamgarh (17) are linked with Chhattisgarh.

We observed bottlenecks in a few corridors, pass through agricultural areas and human settlements. The likelihood of human-elephant conflict increases in the proximity to these corridors. There are small natural habitats observed between Kapilash-Chandaka (9–18), Garjanpahar-Debrigarh (4–5) and Chandaka-Barbara (18–16), with little food, extensive agricultural lands, proximity to urban areas, and increased human activities. Thus, there is a greater risk of human-elephant conflict. The overlap of natural and anthropogenic habitats gives refuge for Asian elephants to move between habitats with relatively less chance of encountering people and availing agricultural food sources. These linkages with a high likelihood of conflict could serve as ecological traps, resulting in more Asian elephant deaths (Goswami & Vasudev, 2017; de la Torre et al., 2021) and reducing gene flow and genetic diversity (De et al., 2021).

Most of the corridors in the present study hold large proportion of protected forest areas (Reserved forests and Protected forests). Only five corridors retained forest areas of more than 75% of their surface. In contrast, one corridor retained only 4.3% forest area (Similipal-Kuldiha), while most corridors (n = 48) retained forest areas on 30–75% of their surface (mean of 47.87% ± 18.22 SD). As a result, most corridor management actions should include habitat improvement in forest areas and restoring corridors in non-forest areas. Wildlife-friendly land-use practices in non-forest lands could be encouraged by establishing a stronger regulatory framework for subsidizing or incentivizing private landowners (Mariyam et al., 2021; Badola et al., 2021). Recognizing Asian elephants as an umbrella species, we emphasize establishing an ecological network to protect and secure landscapes, promoting sustainable forest use, wildlife management, and habitat restoration, and supporting intelligent and sustainable development in collaboration with local communities.

Our study used detailed records from newspaper databases, alongside direct observations which provided wide coverage of the species in the study area. Although newspaper records are widely used in broad-scale ecological studies, such data come with limitations, such as misidentification of species, and under or over-reporting of cases. Despite such restrictions, newspaper data offer extensive coverage and is a cost-effective source of information. Spatial modelling can benefit from careful interpretation and meticulous data processing, especially when substantial quantities are collated over a wider spatial scale. Our predictive ensemble habitat suitability model may have limitations due to sampling bias towards easily accessible locations. Presence-only models assume equal sampling effort throughout the study area (Yackulic et al., 2013). We have taken into account the limitations by carefully incorporating a large set of occurrence data covering a wider region (Phillips et al., 2009; Bystriakova et al., 2012). Additionally, we have employed spatial filtering techniques for the occurrence data and utilized pseudo-absence data. Despite the potential challenges and limitations, species distribution models (SDMs) have emerged as crucial tools for species conservation and management. The occupancy framework is a reliable method for studying species habitat use as it provides an unbiased estimate of the probability of species presence, considering imperfect detectability. However, the occupancy survey requires multiple surveys and is logistically difficult to cover a large region. Also, the inverse of habitat suitability as a resistance map may introduce biases as it overlooks the complexity of actual movement patterns and other factors influencing species dispersal. Landscape genetics and GPS collar data should always be recommended over habitat suitability models, but these data may be expensive and only sometimes be available.

Conclusions

The present study underscores the critical need for conservation of Asian elephants in the face of human-induced threats. The survival of Asian elephants is jeopardized by habitat modification and fragmentation. Our spatially explicit prediction models identified dense forest areas and undulated terrain as key factors increasing habitat suitability of elephant in Odisha. Additively, 27.9% Odisha’s area was deemed suitable for elephants. Connectivity among the forest patches through high centrality links, especially with habitat bottlenecks, needs to be ensured in order to maintain the safe movement and dispersal.

Given the reliance of elephants on traversing fragmented landscapes, it is imperative to establish, restore, and safeguard forest corridors. This will help in mitigating human-elephant conflicts during dispersal and in reducing the risk to both human and elephants. Effective management and expansion of protected area network are essential to maintain the habitat use and connectivity. Our findings offer valuable insights for long-term elephant-centric landscape conservation, emphasizing the urgency of strategic interventions to safeguard elephant populations in Odisha.

Supplemental Information

Supplemental Information 1 Code

Supplemental Information 2 Supplemental Tables and Figure

We are grateful to the State Forest Department of Odisha for their kind co-operation. We are thankful to Hemanta Kumar Sahu, Pratyush P. Mohapatra and Prasad Kumar Dash for their valuable support during field surveys. AKJ thanks Sree, Mohit Payal, Devendra Pandey and Ritesh Vishwakarma for their helps and suggestions for the manuscript.

Additional Information and Declarations

Competing Interests

Author Contributions

Data Availability

Himanshu Shekhar Palei is a volunteer Research Associate at the Aranya Foundation. The authors declare there are no competing interests.

Himanshu Shekhar Palei conceived and designed the experiments, performed the experiments, prepared figures and/or tables, authored or reviewed drafts of the article, and approved the final draft.

Ashish Kumar Jangid conceived and designed the experiments, performed the experiments, analyzed the data, prepared figures and/or tables, authored or reviewed drafts of the article, and approved the final draft.

Dhamdhere Dhanraj Hanumant performed the experiments, authored or reviewed drafts of the article, and approved the final draft.

Nimain Charan Palei performed the experiments, authored or reviewed drafts of the article, and approved the final draft.

Arun Kumar Mishra performed the experiments, authored or reviewed drafts of the article, and approved the final draft.

The following information was supplied regarding data availability:

The code is available in the Supplemental File.

The location data was available for peer review and is not published to protect the species presence locations. This is intended to safeguard against possible misuse for illegal activities. To access species presence locations for research purposes, individuals must email Dr. Subrat Debata of Aranya Foundation (subrat.debata007@gmail.com). Authorized uses of the data include research, analysis, and decision-making processes aimed at further advancements in models. Applicants are required to furnish concise details outlining the intended use of the dataset and proposed methodologies. Dr. Debata will evaluate the appropriateness of each inquiry, ensuring adherence to ethical and scientific standards. Applicants are strictly prohibited from unauthorized sharing, commercialization, or engaging in any illegal activities with the accessed data. Regardless of use, ownership of the location data will remain collectively with all the authors of the article.

Once access is granted, applicants are encouraged to utilize the location data for inference-based research aimed at understanding species ecology and conservation. Proper citation of the publication (i.e., Palei et al., 2024) is required when using the data, and applicants are strictly prohibited from using the location data without providing this citation. Access to the data will be granted within one month of the request.

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
