# Peer review of "On the elephant trails: habitat suitability and connectivity for Asian elephants in eastern Indian landscape"

_PeerJ, doi:10.7717/peerj.16746_

## Round 0.1 · original submission · Major Revisions

Dear Authors,

I extend my heartfelt gratitude for choosing PeerJ as the potential platform for your captivating publication. Following a meticulous evaluation by three esteemed reviewers, I wish to share their insightful feedback with you. While two reviewers recommended minor revisions, a third pointed out the need for more substantial changes. Consequently, I have initiated the process of significant corrections, focusing primarily on refining the materials and methods, enhancing the clarity of raw data presentation, and improving the overall quality of the English language used in the manuscript.

Your dedication to these revisions is sincerely appreciated, and I eagerly anticipate the improved version of your work.

Warm regards,

Armando Sunny

**Language Note:** The Academic Editor has identified that the English language must be improved. PeerJ can provide language editing services - please contact us at copyediting@peerj.com for pricing (be sure to provide your manuscript number and title). Alternatively, you should make your own arrangements to improve the language quality and provide details in your response letter. – PeerJ Staff

Reviewer 1 ·

Basic reporting

This is an interesting manuscript that investigates location and assesses key areas for the movement of Asian elephants. The manuscript is clearly structured as per the PeerJ author guidelines, and is well supported with a range of relevant references. In some cases, however, the relevant literature is not cited, and as such the source of information is not clear. There is also some ambiguity regarding some of the language used: I would therefore recommend a full proof read to reduce the number of grammatical errors in the work.
Many figures are provided and they are clear and professionally formatted. My one recommendation for the figures is to make sure that the scales are always made clear (so that they are easy to understand for readers without reference to other areas of the work).
Raw data have not been provided: only the supplementary materials are included alongside the manuscript. These raw data therefore should be provided alongside the submission.

Experimental design

The aims of this study are clearly stated, with a sound rationale developed. The knowledge gaps is clearly indicated at the end of the introduction, and the wider question fits well within the wide scope of PeerJ's offerings..
While it appears that an extensive dataset has been developed, there seems to be rather patchy distribution of data, with some potential biases towards convenience sampling. It is difficult to evaluate this in greater depth as the data files have not been provided alongside the manuscript. There seem to be different years of data for different data collection types (e.g. newspaper sightings do not match the remainder of the data, and there is a single reference to previous literature, which is not fully explained).
In short, the data shows evidence of convenience sampling and bias (through the selection of newspapers) and these will have influenced the results that have been generated. While it is unlikely that the authors can generate a more holistic dataset at this stage, there needs to be more transparency on the types of data inputted, the raw data files, and the likely influences of convenience sampling on the results.

Validity of the findings

The work uses modelling to generate some interesting and novel findings, and to highlight some areas that could be focused on in terms of conservation. However, the patchiness of the data and the lack of clarity regarding some of the methods mean that the chances of limitations is limited. There is some good justification for choices of test and the outputs are well reported. However, the effects of sampling methods need to be acknowledged more clearly in the results and discussion - otherwise the conclusions are not likely to be reflective of the actual data.
The conclusion is currently far too long and does not reflect the findings of the study. Please reduce the conclusion to 1-2 brief paragraphs that summarise the work. There should be a few more paragraphs in the discussion that explore the work's limitations and the future directions of study.
There also needs to be much clearer acknowledgment as to what the results mean. This is especially true with regards to ruggedness. At current it is suggested that elephants may prefer rugged habitats. Please make the point clearer that it is likely that rugged areas are less likely to be habitated for people so are available to elephants.

Additional comments

Dear Authors,
Thank you for submitting this manuscript that explores elephant habitat. This is an interesting study but there needs to be some work on the manuscript. Please address the following areas:
1. Grammar - ensure the work is proof read in full
2. Provide raw data
3, Ensure the data collection methods are explained at a level that allow replication of results.
4. Please evaluate the impact of convenience and bias on the findings.

Annotated reviews are not available for download in order to protect the identity of reviewers who chose to remain anonymous.

Reviewer 2 ·

Basic reporting

The manuscript is clearly written in appropriate scientific style. The literature cited is comprehensive that gives sufficient background information.

The article structure is good, figures and tables are appropriate, and supplementary data are acceptable.

The results of the study conforms to the objectives mentioned.

Experimental design

No comment

Research question is well defined, relevant and meaningful. It fills an identified knowledge gap.

Technical standards are high although I am not qualified to evaluate it in greater detail as I do carry out this type of modelling

Methods are appropriate. However correcting a bias by shifting elephant locations to forest areas (from L193) when conflicts occur in villages may alter the final model in ways that cannot be controlled. If some studies indicate that human-elephant conflict incidents are often associated with the proximity of human settlements to forested areas, elephants also move in parts of human dominated landscape because the terrain is easier, for water, food and dispersal.

Validity of the findings

The findings are important for the conservation of the Asian elephant metapopulation structure in Odisha as it identifies potential linkages, core areas and pinch points between reserves that can then be protected and restored for ensuring connectivity.

The conclusions are too long and matter pertaining to methods can be incorporated into the Methods section. Conclusions should be a wrapping up of the major findings of this study in the context of other studies, and its applications for conservation of the species.

Reviewer 3 ·

Basic reporting

no comment

Experimental design

If all analysis was done at 1-km resolution, please add a sentence to clarify if the analysis was conducted exclusively at a 1-km resolution.

Page 8 (198-214): Could you provide a more detailed explanation of the process for generating random points? Were these random points specifically generated in relation to conflict points, or were they created independently of any existing data? It seems like the selection of these points was based on a 10 km buffer around the actual occurrence points, but a clearer confirmation of this would be helpful for better understanding.

Page 9 (218-243): Please provide more information on the pre-processing of spatial data, particularly for datasets like bioclim and digital elevation models, which may have resolutions differing from the 1-km resolution. Which method was used to represent these data at 1-km scale.

Page 10 (252-255): I would appreciate a more in-depth clarification regarding the generation process of pseudo-absent data.

Validity of the findings

no comment

---

## Round 0.2 · Minor Revisions

Dear Authors,

The manuscript has undergone significant enhancements, marking commendable progress. Nonetheless, the reviewers suggest that minor corrections are still required to align the manuscript for acceptance and eventual publication. I express my sincere gratitude for your diligent efforts in implementing the suggested improvements. With optimism, I anticipate the swift acceptance of the manuscript for publication.

Warm regards,

Armando Sunny

Reviewer 1 ·

Basic reporting

The initial concerns regarding the formatting of the work have been carefully addressed. The work is now clearer to read and the focus of the work is more transparent throughout the manuscript. The revised conclusion is clearer and the raw data has now been provided.

Experimental design

While there remains a sources of bias in terms of the literature, this is now better acknowledged and evaluated in the discussion. The methods are clearly explained.

Validity of the findings

While there remains evidence of convenience sampling, the authors have justified their reasoning for this well. The conclusion is now much clearer.

Reviewer 2 ·

Basic reporting

The language and style have improved considerably. References are sufficient and the structure, figures and tables are alright. Raw data is shared but on the condition that it is not made public for reasons that I agree with, as the species concerned is Endangered.

Experimental design

The necessary points raised by the reviewers have been addressed.

Validity of the findings

The Conclusion section has been revised, but its repetitive as the technical aspects have already been presented. The authors should present it as a conservation paper rather than focus on the technical aspect. It completely misses the main point that these elephants are surviving in a highly fragmented landscape and have to traverse human dominated areas for mating and dispersal that results in death of many people and of elephants. Therefore its vital that corridors are identified, restored and protected so that croplands are avoided during their movement and dispersal.; and protected areas are well managed and expanded thereby reducing contact with humans.

---

## Round 0.3 · accepted · Accept

Dear Authors,

I am pleased to inform you that both reviewers have concurred on the appropriateness of the corrections implemented in your manuscript. As a result, it is with great satisfaction that I announce the acceptance of your work for publication in PeerJ. I extend my heartfelt gratitude for the meticulous attention to detail demonstrated in your revisions.

Thank you for selecting PeerJ as the platform for sharing your compelling and valuable research. Your contribution is integral to advancing our understanding and promoting the conservation of this species.

Warm regards,

Armando Sunny

Reviewer 2 ·

Basic reporting

The necessary changes have been made

Experimental design

Acceptable

Validity of the findings

Very useful for landscape management of elephant populations

Additional comments

I also recommend that APC charges be waived as the study is good and useful for conservation. As is the situation for many conservation biologists in India, we don't have the funding to pay for APC charges.

Reviewer 3 ·

Basic reporting

-

Experimental design

The manuscript now includes explicit statements confirming the exclusive use of a 1-km resolution and a detailed explanation of the random points generation process. These revisions substantially strengthen the manuscript.

Validity of the findings

-

Additional comments

-